# Eicosapentaenoic Acid and Urolithin a Synergistically Mitigate Heat Stroke-Induced NLRP3 Inflammasome Activation in Microglial Cells

**DOI:** 10.3390/nu17193063

**Published:** 2025-09-25

**Authors:** Hyunji Cho, Judy Kim, Yongsoon Park, Young-Cheul Kim, Soonkyu Chung

**Affiliations:** 1Department of Nutrition, University of Massachusetts, Amherst, MA 01003, USAjudykim@umass.edu (J.K.); yckim@nutrition.umass.edu (Y.-C.K.); 2Department of Food and Nutrition, Hanyang University, Seoul 04763, Republic of Korea; yongsoon@hanyang.ac.kr

**Keywords:** EPA, urolithin A, heat stroke, microglial cells, neuroinflammation

## Abstract

**Background/Objectives**: Global warming and concomitant extreme weather events have markedly increased the incidence of heat stroke. Heat stroke (HS) poses a substantial threat to cerebral health by triggering neuroinflammation and accelerating neurodegenerative processes. The activation of the Nod-like receptor protein 3 (NLRP3) inflammasome for interleukin-1β (IL-1β) secretion has been implicated as a critical mechanism underlying HS-related fatalities. However, the potential role of specific dietary factors to counteract heat stroke-induced neurotoxicity remains largely underexplored. We previously reported that eicosapentaenoic acid (EPA) and urolithin A (UroA), a gut metabolite of ellagic acid, effectively suppress NLRP3 inflammasome activation against metabolic or pathogenic insults. This study aimed to assess the impact of eicosapentaenoic acid (EPA), urolithin A (UroA), and their combination on mitigating heatstroke-mediated NLRP3 inflammasome activation in microglial cells. **Methods**: In vitro heatstroke conditions were replicated by subjecting murine BV2 microglial cells to a high temperature (41 °C) under hypoxic conditions. To achieve nutrient loading, BV2 cells were preincubated with either EPA (50 µM) or UroA (10 µM). NLRP3 inflammasome activation was evaluated by proinflammatory gene expression, caspase-1 cleavage in cells, and IL-1β secretion to the medium. The caspase-1 activation was determined using a luciferase-based inflammasome and protease activity reporter (iGLuc) assay. **Results**: Exposure to high temperatures under hypoxia successfully mimicked HS conditions and promoted NLRP3 inflammasome activation in BV2 cells. Both EPA and UroA substantially attenuated the heat stroke-induced priming of proinflammatory genes. More importantly, EPA and UroA demonstrated a synergistic effect in mitigating HS-induced active caspase-1 production, leading to a dramatic decrease in IL-1β secretion. This synergistic effect between EPA and UroA was further confirmed by the iGLuc reporter assay. **Conclusions**: Dietary enrichment with EPA and UroA precursors may constitute an efficacious strategy for mitigating heat stroke-mediated neuroinflammation and neurodegenerative diseases.

## 1. Introduction

According to the World Health Organization (WHO), intensifying global warming has led to an increase in the frequency, duration, and severity of extreme temperature events [1]. Between 2000 and 2016, the number of people experiencing heatwaves globally increased by around 125 million, and this number is continuously growing [1]. The WHO reported that approximately 489,000 heat-related deaths occurred per year between 2000 and 2019 [2]. Also, based on data from the Centers for Disease Control (CDC), the heat-related mortality rate has been increasing dramatically since 2016 [3]. It is noteworthy that advancing age and chronic inflammatory conditions, including obesity, type 2 diabetes mellitus, dementia, and cardiovascular disease, markedly elevate susceptibility to extreme heat-related morbidity and mortality [4]. Heat stroke (HS) is a severe form of heat-related illness characterized by a body temperature rising to 40 °C (104 °F) or higher, resulting from the body’s inability to dissipate heat effectively. HS rapidly impairs the function of the brain and lungs, manifesting as headache, shortness of breath, slurred speech, confusion, delirium, seizure, and coma. Immediate cooling is an essential treatment to prevent further systemic organ failure [5]. Given that HS is a life-threatening medical situation, it is important to develop proper intervention strategies to prevent or attenuate HS-mediated cerebral damage.

The NLRP3 inflammasome is a key component of innate immune responses, functioning as a multiprotein cytosolic complex that detects pathogen-associated molecular patterns (PAMPs) or damage-associated molecular patterns (DAMPs), leading to the production of proinflammatory cytokines of interleukin (IL)-1β and IL-18 [6]. Recent studies have revealed that NLRP3 inflammasome activation is one of the underlying mechanisms of HS in linking cerebral dysfunction [7,8]. In support of this notion, IL-1β release from microglial cells can induce neuronal injury or pyroptotic cell death, contributing to the onset of neurodegenerative diseases such as Alzheimer’s disease (AD) and multiple sclerosis [9]. These studies suggest that inhibiting the NLRP3 inflammasome is a critical target for preventing or attenuating HS-associated neurological damage [6]. Several dietary factors are known to reduce the activation of the NLRP3 inflammasome, including n-3 polyunsaturated fatty acids (PUFAs) [10], various polyphenols [11], vitamins [12,13,14,15], and dietary fibers [16]. However, direct evidence linking dietary factors to suppression of HS-mediated inflammasome activation in the brain is limited.

In this study, we hypothesize that anti-inflammatory dietary factors suppress HS-induced NLRP3 inflammasome activation in the brain. We aimed to evaluate the potency of two dietary factors, i.e., eicosapentaenoic acid (EPA) and urolithin A (UroA), in mitigating NLRP3 inflammasome activation in response to HS in microglial cells, the resident macrophages in the brain. The rationale for choosing EPA and UroA is to leverage our previous research that both dietary factors exert strong anti-neuroinflammatory effects, albeit through distinct mechanisms of action [17,18]. Using the in vitro setting for heatstroke in BV2 microglial cells, we demonstrated that EPA and UroA effectively dampen the HS-mediated IL-1β secretion by synergistically blocking both priming and assembly of the NLRP3 inflammasome. Considering the growing public health threat posed by extreme heat events, our work offers timely and valuable insights into food-derived dietary factors that can mitigate the impact of extreme heat events.

## 2. Materials and Methods

### 2.1. Cell Culture and NLRP3 Inflammasome Activation

Murine BV2 microglial cells were initially purchased from ACCegene and provided by Dr. Janos Zempleni at the University of Nebraska, Lincoln, NE, USA. BV2 cells were cultured in high glucose Dulbecco’s minimal essential medium (DMEM), containing 4.0 mM L-glutamine without sodium pyruvate (Gibco; Thermo Fisher Scientific, Waltham, MA, USA) supplemented with 10% fetal bovine serum (Phoenix Scientific, Saint Joseph, MO, USA), penicillin (100 IU/mL) and streptomycin (100 µg/mL) (HyClone, Logan, UT, USA). BV2 cells were maintained in a humidified atmosphere at 37 °C and 5% CO_2_, with the medium changed every other day. The seeding density was 1 × 10^5^ cells per 6-well plate.

BV2 cells were pretreated with either 50 µM EPA (NuChek Prep, Elysian, MN, USA), 10 µM UroA (Chengdu Biopurify Phytochemical, Chengdu, Sichuan, China), or a mixture of both for 48 h. The concentrations of EPA (50 µM) and UroA (10 µM) were selected based on the physiologically attainable levels in cerebral circulation, where they are known to exert anti-inflammatory responses [19,20]. To induce PAMP-associated NLRP3 inflammasome, BV2 cells were primed with 50 ng/mL of lipopolysaccharide (LPS) for 1 h and subsequently subjected to 5 µM nigericin (Ng) (Millipore Sigma, Burlington, MA, USA), a bacterial toxin and K^+^/H^+^ ionophore facilitating K^+^ efflux, for 1 h, as modified with our previous study [21].

### 2.2. In Vitro Modality of Heat Stroke (HS)

To simulate cerebral HS conditions in vitro, BV2 cells were replaced with a new medium that was deoxygenated by N_2_ purge and preheated to 41 °C. BV2 cells were transferred to a portable hypoxic chamber (~1% O_2_) and then incubated in a humidified atmosphere at 41 °C for 1 h. To further simulate HS conditions with existing inflammation, BV2 cells were first stimulated with LPS (50 ng/mL) for 3 h, followed by HS stimulation for 1 h.

### 2.3. MTT Assay

The cytotoxic effect of extreme HS was determined using the MTT cell viability kit (Invitrogen, Carlsbad, CA, USA) according to the manufacturer’s protocol.

### 2.4. iGLuc Reporter Assay

The J774 macrophages (Mϕ) stably transfected with the pro-IL-1β-*Gaussia* luciferase fusion (iGLuc) construct were a generous gift from Dr. Hornung at the University of Bonn (hereafter referred to as iJ774 Mϕ, Figure 1B) [22]. iJ774 Mϕ were cultured under the same conditions as BV2 cells. To determine *Gaussia* luciferase activity, the *Gaussia* luciferase glow assay kit (Thermo Fisher Scientific) was used and read with a Synergy H1 multi-mode reader (BioTek, Winooski, VT, USA), as we had conducted previously [21].

### 2.5. Real-Time Quantitative PCR

Total RNA was isolated using the TRIzol Reagent (Invitrogen). RNA concentrations were measured using a Synergy H1 hybrid plate reader (BioTek), and 1 µg of mRNA was reverse-transcribed to cDNA in a total volume of 20 µL using iScript cDNA Reverse Transcription Supermix (BioRad, Hercules, CA, USA). SYBR Green (BioRad, Hercules, CA, USA) supermix was used for RT-qPCR. The 2^−ΔΔCT^ method was used to determine the relative expression with normalization by the housekeeping gene, hypoxanthine-guanine phosphoribosyl transferase (*Hprt*). The primer sequences are available in the Appendix A.

### 2.6. Western Blot Analysis and ELISA

Cell extracts were prepared in RIPA buffer containing protease and phosphatase inhibitors (Millipore Sigma, Burlington, MA, USA), and total protein concentrations were determined using the Pierce BCA Protein Assay Kit (Thermo Fisher). Proteins (20 µg) were separated on 12.5% polyacrylamide gels, transferred to polyvinylidene fluoride membranes, and blocked for 1 h at room temperature with 5% skim milk in Tris-buffered saline with 0.1% Tween 20. The membrane was incubated with primary antibodies (Abcam, Cambridge, UK) against the cleaved caspase-1 (1:500), IκB-α (1:1000), and β-actin (1:1000) in 10% casein overnight at 4 °C and horseradish-peroxidase-conjugated secondary antibody anti-rabbit IgG (1:2000) with 5% skim milk for 1 h at room temperature. Immunoreactive bands were scanned using the Odyssey Infrared Imaging System (LI-COR, Lincoln, NE, USA) to estimate the total protein per lane, and β-actin was used as a normalization control.

For the detection of IL-1β secretion, cell culture supernatants were collected and centrifuged (10,000× *g*, 5 min) to remove cell debris. IL-1β level in cell culture supernatants was quantified by using an ELISA kit (R&D System, Minneapolis, MN, USA) using a Synergy H1 hybrid plate reader (BioTek) according to the manufacturer’s protocol.

### 2.7. Statistical Analysis

All values were expressed as mean ± SEM. Student’s *t*-test was used for comparison between the two groups. Multi-group comparisons were performed by a one-way ANOVA followed by Tukey’s multiple comparison test, and the differences were considered significant at *p* < 0.05. To generate a heat map of the proinflammatory gene profile, qPCR values for each gene were standardized and expressed as Z-scores using pooled samples (*n* = 8). All statistical analyses were performed with GraphPad Prism (Version 9.1.2., GraphPad Software, Inc., San Diego, CA, USA).

## 3. Results

### 3.1. EPA and UroA Synergistically Suppressed the PAMP-Induced NLRP3 Inflammasome Activation in BV2 Cells

The activation of the NLRP3 inflammasome comprises two separate steps: (1) priming of necessary genes for the inflammasome components, such as *Nlrp3*, *Il1β*, and *Il18*, and (2) assembly of the inflammasome for caspase-1 activation. We first tested whether EPA, UroA, and their combination exert distinct mechanisms in suppressing NLRP3-inflammasome in the microglial BV2 cells. After treatment with LPS for priming, BV2 cells were stimulated with Ng, a bacterial toxin and a very well-established PAMP signal. As expected, priming of the inflammasome was significantly suppressed by individual treatment with EPA and Uro A, which decreased the expression levels of *Il1β* and *Nlrp3*. However, the synergistic effect between EPA and UroA in decreasing the priming steps was not observed (Figure 1A). Next, we tested whether EPA and UroA could inhibit inflammasome assembly and caspase-1 activation. For this, we employed the iGLu reporter assay by stimulating LPS plus Ng (Figure 1B). Treatment of each EPA and UroA decreased *Gaussia* luminescence by half (Figure 1C). More importantly, combined treatment with EPA and UroA exhibited a potent synergistic effect in suppressing *Gaussia* luminescence, almost completely dampening it to levels comparable to those of the control. Taken together, our findings indicate that EPA and UroA each effectively attenuate the priming phase of the inflammasome pathway without exhibiting synergistic effects. In contrast, their combined administration synergistically suppresses NLRP3 inflammasome assembly, thereby inhibiting caspase-1 activation in response to pathogenic stimuli.

### 3.2. Establishment of Heat Stroke Modality In Vitro Using BV2 Microglial Cells

We aimed to establish in vitro conditions of HS in BV2 cells, as a tool to screen inhibitory factors. We used reduced cell viability as the overt sign of HS in response to high temperatures and low oxygen levels, which reflects hypoxic conditions in the brain due to reduced respiratory capacity at high temperatures. Neither high temperature (41 °C) nor hypoxic conditions (~1% O_2_) for one hour reduced cell viability compared to the control BV2 cells grown at 37 °C. Exposure to 41 °C under hypoxic conditions for 1 h significantly decreased cell viability of BV2 cells (Figure 2A). Given this result, we defined simultaneous stress of 41 °C and hypoxia (~1% O_2_) as heatstroke conditions (HS). Next, we determine whether this HS could replicate the NLRP3 inflammasome activation. The HS alone significantly increased the *Il1β* and *Nlrp3* compared with control BV2 cells (Figure 2B). To further simulate the metabolic susceptibility and preexisting inflammatory conditions, BV2 cells were first treated with LPS, followed by HS for 1 h. Prior treatment of LPS dramatically increased HS-induced *Il1β* and *Nlrp3* compared to HS only (Figure 2B).

In summary, by combining high temperature and hypoxic stresses in BV2 cells, we successfully established the in vitro heatstroke model of the brain, reproducing NLRP3 inflammasome activation and reduced cell viability. We also validated that this modality could represent metabolically vulnerable conditions by incorporating inflammatory stimulation prior to HS induction.

### 3.3. EPA, UroA, and Their Combination Were Effective in Preventing HS-Induced Cell Viability by Mitigating NLRP3 Inflammasome Activation

By leveraging the HS modality that we established in Section 3.2, we next asked whether EPA and UroA, individually or in their co-treatment, could attenuate HS-induced damages in the brain. In response to 1 h of HS, only ~60% of cells were viable. In contrast, pretreatment with EPA (50 μM) and UroA (10 μM) significantly increased cell viability to 75% and ~85%, respectively. Notably, the combination of EPA and urolithin A almost completely blocked HS-induced cell death, comparable to untreated BV2 cells (Figure 3).

Given that EPA and urolithin A were capable of rescuing the HS-mediated cell death (Figure 3), we next questioned whether EPA, UroA, or their combination mitigates HS-induced NLRP3 inflammasome activation, even in conditions of preexisting inflammation. To address this hypothesis, BV2 cells were first cultured with supplementation of EPA, UroA and their combination, then employed HS induction with or without LPS stimulation. The HS alone increased the mRNA expression of *Il-1β* by ~5-fold, which was significantly attenuated by supplementation of EPA, UroA and their combination in a similar potency (Figure 4A). Similarly, HS induced *Nlrp3* expression by ~2.5-fold, which was reduced by UroA but not by EPA alone (Figure 4B). In the presence of LPS which models preexisting inflammation conditions, HS-induced *Il-1β* and *Nlrp3* expressions were considerably higher than controls of HS alone (Figure 4B, shaded column). Notably, both EPA and UroA significantly decreased the expression of *Il-1β* and *Nlrp3*. Especially, UroA dampened HS-associated proinflammatory gene expression comparable to LPS-only control (Figure 4A,B, shaded column). These results demonstrated that both EPA and UroA are effective in reducing the priming of the inflammasome, although the synergistic effect in the priming step is uncertain. Next, we further investigated the impact of EPA and UroA on HS-mediated inflammasome assembly for caspase-1 activity and IL-1β secretion. The HS alone significantly increased IL-1β secretion, which was further increased by >2-fold in BV2 cells pretreated with LPS (Figure 4C). Supplementation with individual EPA and UroA substantially decreased the IL-1β secretion and, more importantly, EPA and UroA showed a synergy to block the IL-1β secretion comparable to untreated cells (Figure 4C, shaded column). To further confirm the role of EPA, UroA, and their combination on caspase-1 activity, we performed the caspase reporter assay using the J774 MΦ hovering iGLuc construct shown in Figure 1C. The combined treatment of EPA and UroA almost completely blocked caspase-1 activation induced by pathogenic invasion (Ng plus LPS), and abolished the HS-induced caspase-1 activation, even in preexisting inflammatory conditions (Figure 4D). Next, we examined the broader cytokine and chemokine gene expression to establish a comprehensive inflammatory profile in response to HS (Figure 4E). Notably, the majority of proinflammatory marker genes—including interleukin-6 (*Il6*), tumor necrosis factor-α (*Tnfa*), inducible nitric oxide synthase (*iNOS*), monocyte chemoattractant protein-1 (*Mcp1*), and cyclooxygenase-2 (*Cox2*)—were markedly upregulated under HS+LPS conditions. In contrast, M2 macrophage markers, including *Il4* and *Il13*, did not exhibit this pattern, whereas *Il16* was significantly decreased by HS irrespective of supplementation.

To validate whether EPA and UroA synergistically block the inflammasome assembly process, we investigated the NFκB activation and caspase-1 cleavage. Supplementation of EPA and UroA each significantly attenuated IκBα degradation, a key mechanism for NFκB nuclear translocation for transcriptional activation of inflammasome components, while their combination of EPA and UroA did not show additional suppression of IκBα degradation (Figure 5).

The cleavage of inactive procaspase-1 via the assembled inflammasome is indispensable for IL-1β secretion [23]. The supplement of EPA and UroA not only individually reduced the caspase-1 cleavage against HS+LPS stimuli but also showed a synergy in virtually complete inhibition of caspase-1 cleavage comparable to the unstimulated sample (Figure 5). Taken together, these results demonstrated that EPA and UroA are similarly effective in mitigating HS-induced NFκB activation and caspase-1 cleavage. It is important to note that combined treatment of EPA and UroA can virtually block the HS-induced IL-1β by synergistically suppressing inflammasome activation for caspase-1 cleavage.

## 4. Discussion

In the present study, we first aimed to establish the in vitro modality for HS and to determine the preventive potential of dietary factors of EPA and UroA against the HS-induced NLRP3 inflammasome activation. The combined stresses of elevated temperature and hypoxia effectively reproduced HS conditions in microglial cells, providing a valid in vitro model to demonstrate the exacerbated severity of HS under preexisting inflammatory states (Figure 2). Our findings further demonstrate that EPA, UroA, and their combination suppress NLRP3 inflammasome activity through a two-pronged mechanism: (1) inhibition of inflammasome priming and (2) impediment of inflammasome activation, acting synergistically (Figure 6). To our knowledge, this is the first study to investigate the individual and synergistic effects of EPA and UroA in attenuating HS-induced NLRP3 inflammasome activation, demonstrating efficacy even under conditions of preexisting inflammation.

Heat-related mortality has been identified as one of the key climate extremes posing a risk to human health and has increased rapidly [24]. Additionally, people aged over 60 years, those with obesity, cardiovascular disease, pulmonary disease, or diabetes are vulnerable to heat waves because of physiological impairments in regulating the core body temperature [25]. Zhang et al. [26] showed that HS decreased the survival rate and increased the serum IL-1β level, which were further aggravated by LPS treatment. Mechanistically, HS is known to induce NLRP3 inflammasome [8], which is a two-step process, priming and activation [27,28]. Once LPS binds to Toll-like receptor, the priming step starts with upregulated expression of the inflammasome components, nuclear factor-κB (NFκB), NLRP3, IκB-α degradation, pro-caspase-1, and pro-IL-1β. NLRP3 inflammasome assembly is activated in pathogenic infection (i.e., bacterial, viral and fungal infections) or other damage-associated molecular patterns (saturated fatty acids, cholesterol, ceramide, uric acid, etc.), which promote assembly of the inflammasome for cleavage of caspase-1 to facilitate the release of IL-1β, toxicity to the central nervous system resulting Parkinson’s disease (PD) and AD [27,29]. In addition to these pathogenic and physiological threats, environmental stress such as HS induced NLRP3 inflammasome in both priming and activation steps, increasing the gene or protein expression of caspase-1, IL-1β, and NLRP3 in cell lysate, and IL-1β in supernatant [30,31,32,33].

In this study, using our in vitro HS modality, we evaluated whether EPA and UroA can exert neuroprotective effects against HS. EPA is not only one of the most effective anti-inflammatory dietary molecules but also a component of the cell membrane. Supplementation of n-3 PUFA increases the EPA levels in the brain and improves cognitive function [34]. It is also supported by multiple studies that EPA suppresses NLRP3 inflammasome activation in microglial cells in response to pathogenic signals [19,35] and in the brain of the animal models, including AD [36,37], cerebral infarction mice [35], and depression [38]. On the other hand, UroA has emerged as a novel player to attenuate neuroinflammations. UroA is a gut metabolite of ellagic acids, abundantly found in pomegranates, strawberries, and walnuts, and known to cross the blood–brain barrier [39]. Extensive studies, including our research [20], have demonstrated the anti-inflammatory function of UroA in the microglial cells [40,41,42] and animal models of neuroinflammation due to aging [43], sleep deprivation [44], and PD [40]. Examination of the combined effects of EPA and UroA is important, since consumption of several nuts, such as walnuts and pecans, can simultaneously produce EPA and UroA. Walnuts are an excellent source of α-linolenic acid (ALA) and ellagic acid, which are endogenously converted into EPA and UroA, respectively. Based on accumulating evidence that walnut exerts neuroprotective effects [45,46,47], it is feasible to assume that the combination of EPA and UroA synergistically inhibits NLRP3 inflammasome activation.

The concentration of EPA in our study was determined based on the previous study showing the beneficial effect of EPA on the NLRP3 inflammasome in BV2 cells primed with LPS [19]. Additionally, Abu et al. [48] showed that the EPA concentration of 50 µM in the present study is reachable in healthy humans. Consistent with the present study, previous in vitro studies showed that the treatment of EPA decreased the expression of NLRP3 and IL-1β in cell lysate with LPS [19,49], and expression of IL-1β and cleaved caspase-1 in cell supernatant with oxygen-glucose deprivation [35], LPS [19], and LPS plus Ng [50], proving that EPA alleviates NLRP3 inflammasome activation as well as priming in vitro. Furthermore, treating the BV2 cells’ medium with LPS decreased SH-SY5Y cell viability, whereas treating it with LPS and EPA increased SH-SY5Y cell viability, suggesting that EPA can protect against neurodegeneration and the NLRP3 inflammasome [49]. Previous in vivo study showed that supplementation of EPA decreased the expression of pro-caspase-1, NLRP3, pro-IL-1β, cleaved caspase-1, and IL-1β, and the level of IL-1β in blood or brain of rodents with depression [38], AD [36,37], and acute cerebral infarction [35], proving that EPA alleviates NLRP3 inflammasome activation as well as priming in vivo. However, EPA has not been studied for protecting HS-mediated neuronal damage and its associated health complications.

UroA is a gut-driven metabolite from ellagic acid that is abundant in pomegranates, strawberries, and walnuts. We used a 10 µM concentration of UroA, which was determined based on the previous study showing the beneficial effect of UroA on the NLRP3 inflammasome in BV2 cells primed with LPS and activated with ATP [20]. Espin et al. [51] showed that 10 µM of UroA concentration is achievable in circulation through UroA-producing diets. Consistent with our study, previous in vitro studies showed that treatment with UroA mitigated the NLRP3 inflammasome activation induced by molecular patterns such as ATP [40,42] or pathogenic threat by bacterial toxin Ng [41]. In support of this in vitro studies, animal studies also showed that supplementation of UroA decreased the brain expression of NLRP3, cleaved caspase-1, and IL-1β, and serum level of IL-1β in mice with osteoporosis [42], aging [43], sleep deprivation [44], and PD [40], proving that UroA is effective in alleviating NLRP3 inflammasome activation in vivo. Additionally, He et al. [52] showed that supplementation of ellagic acid decreased the expression of NLRP3, cleaved caspase-1, and IL-1β, and attenuated dopamine neuronal damage in the brain of rats with PD, implicating that UroA, its major gut metabolite from the ellagic acid, could protect neurons from NLRP3 inflammasome in the brain. However, there has been no study investigating the beneficial effect of UroA on the NLRP3 inflammasome against HS.

Walnuts are known to be a nutrient-dense “brain food” for their high content of unsaturated fatty acids, proteins, polyphenols, and minerals, exerting neuroprotective effects against neurodegenerative diseases, including AD and PD [53]. Walnuts produce UroA as a gut metabolite by microbiota and contain ALA, which converts to EPA endogenously, suggesting that the beneficial effect of walnuts can be regarded as a combination effect of EPA and UroA. Consistent with our results, there is literature support showing that the supplementation of walnuts decreased the expression of NLPR3, caspase-1, and IL-1β in the colon of mice with colitis [54,55]. However, although our results suggest a potential protective role of walnut supplementation, further validation through additional mechanistic studies and clinical investigations is required before emphasizing its translational impact. In addition to EPA, docosahexaenoic acid (DHA) is known to inhibit NLRP3 inflammasome in myeloid-derived suppressor cells (MDSC) [56] and mice with seizures and depression [38]. However, previous studies showed that there are no differences between EPA and DHA on the released IL-1β expression or level, the biomarker for the NLRP3 inflammasome activation, in MDSC supernatant [56] and the brain of mice with seizure and depression [38]. After ingestion, ALA can be endogenously metabolized through desaturation, elongation, and peroxisome oxidation to EPA and DHA with the conversion rate of 5–10% and 2–5% in healthy adults, respectively [57]. In our hands, the beneficial effect of EPA on the gene expression of *Nlrp3* was greater than that of DHA, but no significant differences were observed between EPA and DHA on the gene expression of *IL1β* in BV2 cells following NLRP3 inflammasome activation after LPS priming and Ng stimulation (Appendix A).

It is important to note that our study revealed the synergistic effects of EPA and UroA in mitigating HS-mediated NLRP3 inflammasome activation, evidenced by reduced *Gaussia* luminescence, a reporter assay for caspase-1 activation, and IL-1β release. Given that the combination of EPA and UroA showed a synergistic effect on the caspase-1 reporter assay and IL1β secretion, but not the priming steps for transcription of NLRP3 components, we speculate that the combination of EPA and UroA synergistically blocks NLRP3 inflammasome assembly. It is also noteworthy that NLRP3 inflammasome-induced cell death, so-called pyroptosis [58], is almost completely blocked by the combination of EPA and UroA (Figure 3). More importantly, the combination of EPA and UroA was able to block the IL-1β secretion comparable to the non-treatment control, even in the presence of preexisting LPS conditions. These results are highly promising for mitigating HS-induced neuronal damage and the incidence of HS through dietary intervention, particularly for vulnerable populations.

As neurodegenerative diseases continue to rise and extreme heat exposure poses significant health threats, investigating the potential therapeutics that target pathways of neuroinflammation in extreme heat conditions is imperative. Our findings and those of others support the potential neuroprotective effects of EPA and UroA synergistically, even in preexisting conditions. Our present study has several limitations. The synergistic therapeutic mechanisms of EPA and UroA in the context of heat stroke (HS) have not been demonstrated in vivo, and their direct effects on neurons remain to be clarified. In addition, the brain bioavailability of these compounds represents an important gap. The concentrations used in our in vitro experiments (50 μM for EPA and 10 μM for UroA) are achievable in systemic circulation through dietary supplementation; however, their ability to cross the blood–brain barrier (BBB) is limited, with estimates suggesting that roughly one-tenth of circulating levels may reach brain tissue [59,60]. Both EPA and UroA can cross the BBB and undergo metabolic processing in the brain, which indicates potential involvement in neuroinflammatory regulation. Further in vivo studies are needed to clarify their relevance in central nervous system inflammation. Despite these limitations, to our knowledge, it is the first study to show the therapeutic effect of EPA and UroA synergistically on the NLRP3 inflammasome activation induced by HS in microglial cells.

## 5. Conclusions

In conclusion, consumption of EPA or UroA-producing foods could synergistically provide protective effects against HS-induced neuroinflammation by suppressing NLRP3 inflammasome activation. These findings highlight novel dietary intervention strategies targeting HS-induced neuroinflammation, particularly for vulnerable populations who have preexisting inflammatory conditions. Additional preclinical investigations are required to validate the synergistic protective effects of EPA and UroA in animal models of heatstroke before advancing these interventions to human clinical trials.

## Figures and Tables

**Figure 1 nutrients-17-03063-f001:**
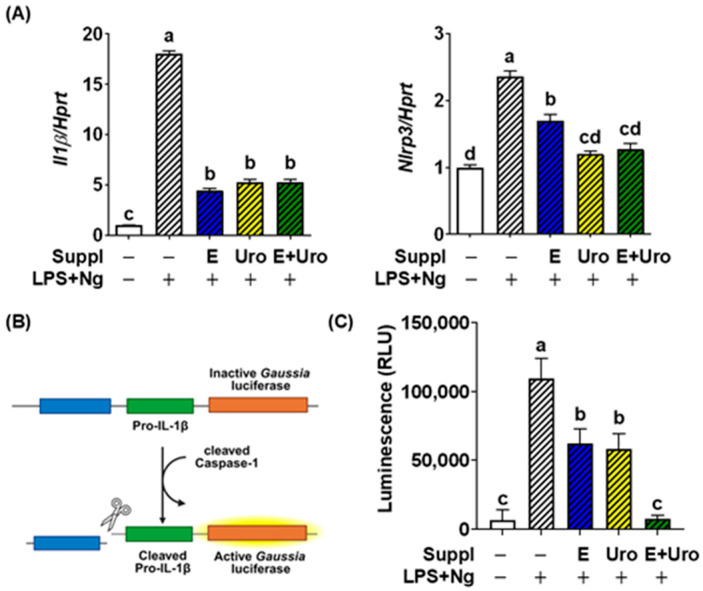
Effectiveness of EPA, UroA, and the combination in mitigating NLRP3 inflammasome activation. (**A**) *Il1β* and *Nlrp3* gene expression, induced by inflammation activation upon LPS plus Ng treatment in BV2 cells. (**B**) Schematic representation of the pro-IL-1β–Gaussia luciferase fusion construct (iGLuc) in J774 macrophages (caspase-1 reporter). Upon inflammasome activation, cleaved caspase-1 promotes pro-IL-1β processing, which in turn triggers activation of Gaussia luciferase. (**C**) Relative *Gaussia* luminescence. All data are presented as mean ± SEM (*n* = 6 per group). Groups not sharing a common letter differ significantly (*p* < 0.05) as determined by one-way ANOVA followed by post hoc Tukey, with values ranked in descending order as a > b > c > d. LPS, lipopolysaccharide; Ng, nigericin; Suppl, Supplementation; E, EPA; Uro, Urolithin A; E + Uro, combined treatment with EPA and Uro.

**Figure 2 nutrients-17-03063-f002:**
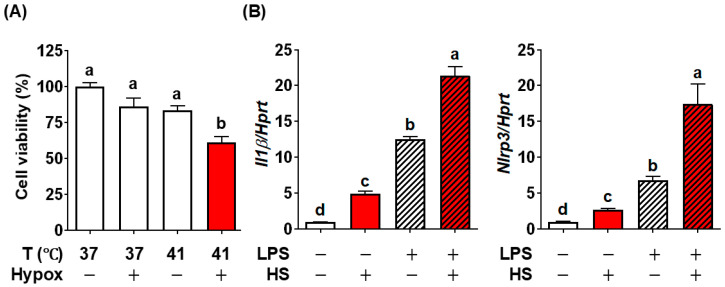
Establishment of HS modality in BV2 cells. (**A**) Cell viability in response to hypoxia and temperature. (**B**) *Il1β* and *Nlrp3* gene expression induced by HS with or without LPS pretreatment. All data are presented as mean ± SEM (*n* = 6 per group). Groups not sharing a common letter differ significantly (*p* < 0.05) as determined by one-way ANOVA followed by post hoc Tukey, with values ranked in descending order as a > b > c > d. HS, heatstroke (41 °C plus hypoxia); Hypox, hypoxia; LPS, lipopolysaccharide.

**Figure 3 nutrients-17-03063-f003:**
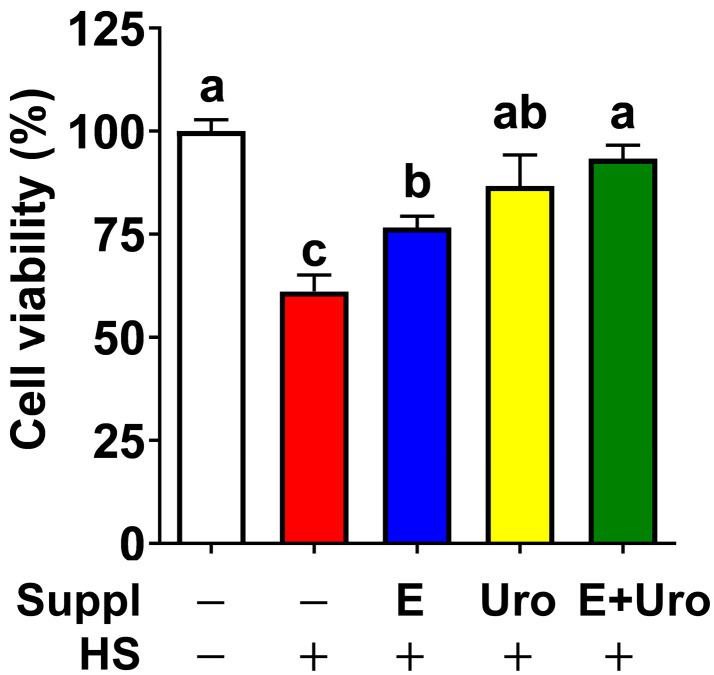
EPA, UroA, and the combination increased cell viability against heatstroke (HS). BV2 cells were cultured with either EPA, urolithin A (Uro) or its combination. Upon induction of HS (41 °C with ~1% O_2_) for 1 h, cell viability was estimated by MTT assay. All data are presented as mean ± SEM (*n* = 6 per group). Groups not sharing a common letter differ significantly (*p* < 0.05) as determined by one-way ANOVA followed by post hoc Tukey, with values ranked in descending order as a > b > c.

**Figure 4 nutrients-17-03063-f004:**
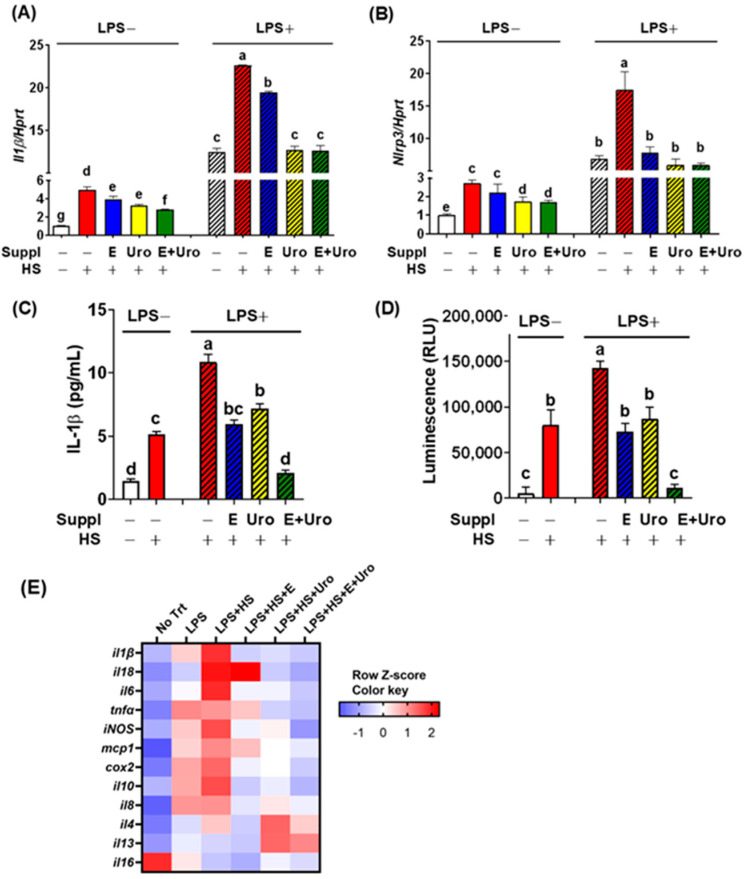
EPA and UroA showed synergy in suppressing HS-induced NLRP3 inflammasome assembly for caspase-1 activation and IL-1β secretion. BV2 cells cultured with either EPA, urolithin or their combination were stimulated with HS or without LPS. (**A**) *Il-1β* gene expression (**B**) *Nlrp3* gene expression (**C**) IL-1β secretion to the medium measured by ELISA. (**D**) Relative *Gaussia* luminescence activity by caspase-1 reporter assay. (**E**) Heat map of proinflammatory gene profile obtained by qPCR (pooled samples of *n* = 8). Values for each gene were standardized and repressed as Z-scores. (**A**–**D**) data are presented as mean ± SEM (*n* = 8 per group). Groups not sharing a common letter differ significantly (*p* < 0.05) as determined by one-way ANOVA followed by post hoc Tukey, with values ranked in descending order as a > b > c > d > e > f > g. Suppl, Supplementation; E, EPA; Uro, Urolithin A; E+Uro, combination of EPA and Urolithin A.

**Figure 5 nutrients-17-03063-f005:**
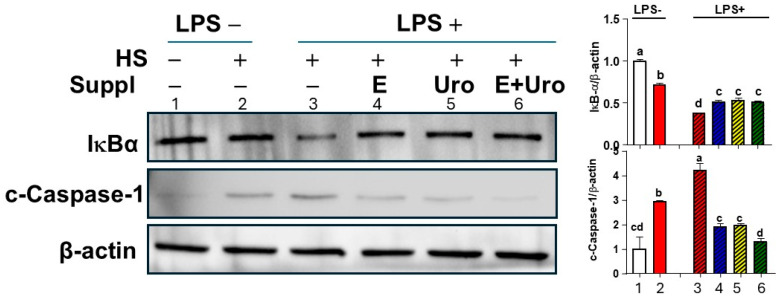
The HS-induced NFκB degradation and caspase-1 cleavage were suppressed by EPA, UroA, and their combination. Western blot images of IκB-α degradation and cleaved caspase-1 and their relative quantification. β-actin as a loading control. Images are representative of three separate experiments. All data are shown as mean ± SEM (*n* = 3). Groups not sharing a common letter differ significantly (*p* < 0.05), as determined by one-way ANOVA followed by Tukey’s post hoc test, with values ranked in descending order (a > b > c > d). E, EPA; Uro, Urolithin A; E+Uro, combination of EPA and Urolithin A.

**Figure 6 nutrients-17-03063-f006:**
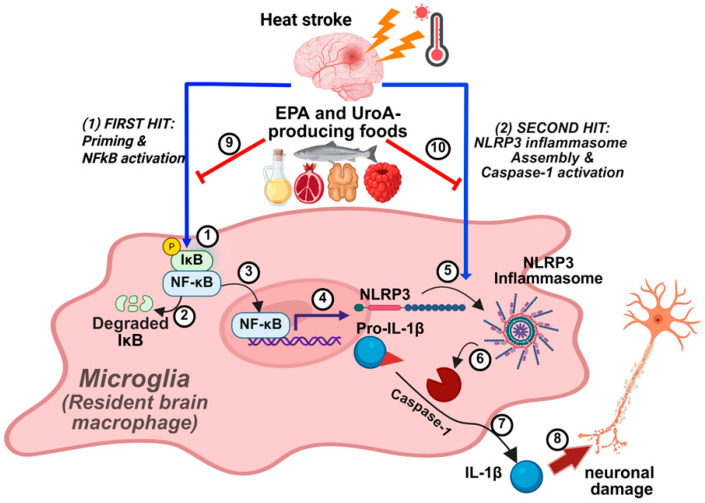
Proposed model of EPA, UroA, or their combination in alleviating heat stroke (HS)-induced neuroinflammation. In microglial cells, HS initiates a “first hit” by activating NF-κB (①) through degradation of IκB protein (②), thereby allowing NF-κB nuclear translocation (③) and priming of proinflammatory genes, including Nlrp3 and pro-Il1β (④). HS further provides a “second signal” that promotes NLRP3 inflammasome assembly (⑤), leading to caspase-1 activation (⑥) and subsequent cleavage of pro-IL-1β, resulting in mature IL-1β release (⑦). The secreted IL-1β, in turn, contributes to adjacent neuronal damage (⑧). Dietary supplementation with EPA or Urolithin A—derived from n-3 PUFA oils, fish, walnuts, pomegranate, or berries—may attenuate HS-induced NF-κB activation (⑨) and block inflammasome activity (⑩), either individually or synergistically.

## Data Availability

The original contributions presented in this study are included in the article/Appendix A. Further inquiries can be directed to the corresponding author.

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
