# Peer review of "Eicosapentaenoic Acid and Urolithin a Synergistically Mitigate Heat Stroke-Induced NLRP3 Inflammasome Activation in Microglial Cells"

_nutrients, 2025, doi:10.3390/nu17193063_

Round 1

Reviewer 1 Report

Comments and Suggestions for Authors

In the article Eicosapentaenoic Acid and Urolithin A Synergistically Mitigate Heat Stroke-Induced NLRP3 Inflammasome Activation in Microglial Cells, the authors analyzed the role of EPA and UroA in the response of microglial cells to heat stroke, finding that both compounds have an anti-inflammatory effect, which does not increase when the two are combined. Below are my suggestions regarding the manuscript:

  1. Regarding the Gaussia luminescence analysis, this assay was performed on a different cell line. Although this is specified in the Materials and Methods section, it should also be clearly stated in the figure legend.
  2. The statistical significance notation should be more clearly explained in the figure legends.
  3. I recommend including additional markers of the inflammatory phenotype of BV2 cells, such as iNOS, by Western blot analysis.
  4. It would be useful to assess the phagocytic capacity of BV2 cells to further demonstrate that EPA and UroA mitigate pro-inflammatory responses in microglial cells.

Author Response:

1. Regarding the Gaussia luminescence analysis, this assay was performed on a different cell line. Although this is specified in the Materials and Methods section, it should also be clearly stated in the figure legend.

In response to the reviewer’s comment, we have revised the figure legend to clearly indicate that the reporter assay was performed in J774 macrophages harboring a caspase-1 reporter, which is distinct from BV2 microglia (brain-resident macrophages).

2. The statistical significance notation should be more clearly explained in the figure legends.

In response to the reviewer’s comment, we have expanded the figure caption to provide a clearer explanation of the statistical significance.

3. I recommend including additional markers of the inflammatory phenotype of BV2 cells, such as iNOS, by Western blot analysis.

We do not currently have an iNOS antibody available and therefore could not obtain Western blot images within the revision deadline. Since iNOS expression is primarily regulated at the transcriptional level, we instead performed qPCR analysis for iNOS together with several other pro-inflammatory genes to provide a more comprehensive assessment. These new results have been incorporated into a heatmap and added as Figure 4E. As Reviewer 3 raised a similar question, we have also addressed this point in our responses to Reviewer 3, where the figure can be found.

4. It would be useful to assess the phagocytic capacity of BV2 cells to further demonstrate that EPA and UroA mitigate pro-inflammatory responses in microglial cells.

We appreciate the reviewer’s valuable insight. At present, several publications demonstrate that UroA promotes mitophagy and enhances microglial plaque clearance; however, no direct studies have evaluated the effects of UroA on efferocytosis. In contrast, there is ample evidence that EPA increases microglial phagocytosis of debris and Aβ, thereby supporting a pro-resolving phenotype. On this basis, it would indeed be intriguing to test whether EPA and UroA exhibit phagocytic effects in our system using fluorescently labeled beads. While such experiments are highly appealing, they represent the next stage of investigation and are beyond the scope of the current study.

See the attached cover letter (response to the reviewers). 

Reviewer 2 Report

Comments and Suggestions for Authors

This is a very interesting original article. It refers to the heatstroke, which has become a problem due to global warming. The Authors aimed to assess the impact of eicosapentaenoic acid (EPA) and urolithin A (Uro A), and their combination on mitigating heatstroke-mediated NLRP3 inflammasome activation in microglial cells. In vitro heatstroke conditions were replicated by subjecting murine BV2 microglial cells to a high temperature and hypoxic conditions. The Authors showed that exposure to high temperature and hypoxia successfully mimicked heatstroke conditions and ptomoted NLRP3 inflammosome activation in BV2 cells. Both compounds substantially attenuated the heatstroke-induced priming of pro-inflammatory genes. A co-application of both compounds led to a synergistic effect in mitigating heatstroke-induced active caspase-1 production resulting in a dramatic decrease in a secretion of Interleukin-1 beta. They concluded that dietary enrichment with precursors of both compounds may constitute an effective strategy for mitigating heatstroke-mediated inflammation and neurodegenerative diseases.

The article can be accepted after a minor revision, Only two minor points should be addressed:

1) Schematic drawing shown in Figure 1B should be presented in more detail to make it more clear especially for non-professional readers.

2) Model proposed in Figure 6 should also be presented in more detail to make it more clear especially for non-professional readers.

Author Response: 

We sincerely appreciate the reviewer’s compliments and encouraging feedback.

  • Schematic drawing shown in Figure 1B should be presented in more detail to make it more clear especially for non-professional readers.

In response to the reviewer’s comment, we have updated Figure 1B and added a more detailed explanation to enhance clarity and improve readability for a lay audience.

  • Model proposed in Figure 6 should also be presented in more detail to make it more clear especially for non-professional readers.

In response to the reviewer’s comment, we have revised the figure by adding more details and numbering the steps to illustrate the sequence of inflammasome activation and the potential targeting points of EPA and UroA. These changes were made to ensure that our proposed model can be more easily understood by a lay audience.

Please see the attached letter for the response to the reviewers.

Reviewer 3 Report

Comments and Suggestions for Authors

This study presents a compelling investigation into the synergistic neuroprotective effects of eicosapentaenoic acid (EPA) and urolithin A (UroA) under heat stroke (HS)-induced stress in microglial cells. Its thorough documentation of the in vitro HS model and integration with prior literature on inflammasome biology underscores the novelty and translational relevance of the findings. The use of complementary modalities—including gene expression, ELISA, luciferase-based assays, and Western blotting—provides a comprehensive overview of the molecular mechanisms underlying inflammasome priming and activation. However, some points merit further consideration:

  1. The study primarily focuses on the anti-inflammatory efficacy of EPA and UroA, yet it does not address their broader therapeutic effects or potential toxicities. Would the inclusion of dose–response curves or cytotoxicity profiles under extended exposure conditions help clarify the safety margins of these compounds?
  2. The paper suggests that EPA and UroA synergistically suppress inflammasome assembly rather than priming. Could further exploration and validation of this hypothesis through additional molecular analyses—such as ASC speck formation, mitochondrial ROS quantification, or NLRP3 oligomerization—strengthen the mechanistic insights regarding inflammasome regulation?
  3. The authors report IL-13 and IL-16 as key inflammatory mediators, yet the role of IL-1β—typically central to NLRP3 activation—is less emphasized. Would a more detailed profiling of cytokine secretion, including IL-1β and IL-18, provide a more complete picture of the inflammatory cascade?
  4. While the study references physiologically relevant concentrations of EPA and UroA, it remains unclear whether these levels are achievable through dietary intake alone. Could the authors elaborate on the pharmacokinetics or bioavailability of these compounds in vivo, particularly in the context of HS or neuroinflammatory conditions?
  5. The proposed dietary implications are intriguing, especially regarding walnut-derived precursors. However, the translational leap from in vitro findings to dietary recommendations may benefit from further substantiation. Could the authors consider framing these suggestions more cautiously or referencing ongoing clinical or preclinical studies?

Author Response:

1. The study primarily focuses on the anti-inflammatory efficacy of EPA and UroA, yet it does not address their broader therapeutic effects or potential toxicities. Would the inclusion of dose–response curves or cytotoxicity profiles under extended exposure conditions help clarify the safety margins of these compounds?

To clarify dose selection and safety considerations: although we did not perform dedicated cytotoxicity assays (e.g., MTT or LDH) for this revision due to rapid revision timeline, we selected 10 µM Urolithin A (UroA) based on our prior work (Toney et al., 2020) and multiple reports indicating that UroA is well tolerated in BV2 cells up to ~30 µM. In the BV2 literature, UroA is commonly used at 2.5–10 µM for mechanistic studies and up to 30 µM for stronger phenotypes without reported cytotoxicity. In our experiments, the combination of 50 µM EPA and 10 µM UroA did not reduce cell viability, as assessed by MTT, even under heat-stress conditions (Figure 3).

Representative references

  • Toney et al., 2020, Int J Environ Res Public Health, 18(1):68 (UroA 10 µM).
  • Velagapudi et al., 2019, Mol Nutr Food Res, 63(10):e1801237 (UroA 2.5–10 µM).
  • Qiu et al., 2022, Neuropharmacology, 207:108963 (UroA 10–30 µM).
  • Mingo et al., 2024, Frontiers in Cellular Neuroscience (UroA 10–30 µM).
  • Chen et al., 2025, Int Immunopharmacology, 48:114057 (UroA 10 µM).

For EPA, prior BV2 work demonstrated no cytotoxicity at 60 µM after 24 h by both MTT and LDH assays; notably, 60 µM EPA was also used during LPS stimulation without loss of viability (Kaohsiung J Med Sci, 2023;39:565–575).

2. The paper suggests that EPA and UroA synergistically suppress inflammasome assembly rather than priming. Could further exploration and validation of this hypothesis through additional molecular analyses—such as ASC speck formation, mitochondrial ROS quantification, or NLRP3 oligomerization—strengthen the mechanistic insights regarding inflammasome regulation?

We sincerely thank the reviewer for this excellent suggestion regarding additional experiments. We fully agree that determination of ASC speck formation would further strengthen the mechanistic basis of our study. At the same time, we would like to emphasize that we have already demonstrated two key findings showing the synergy between EPA and UroA in attenuating inflammasome activation: (i) reduced IL-1β secretion as measured by ELISA, and (ii) decreased caspase-1 cleavage as confirmed by Western blot. These results were further validated using a caspase-1 reporter assay. Taken together, these complementary approaches provide critical evidence supporting our proposed mechanism. Thus, while ASC speck formation would provide an informative supplement, it is not indispensable for establishing the central conclusion.

Nevertheless, in response to the reviewer’s recommendation, we have initiated experiments to evaluate ASC speck formation by immunostaining and/or Western blot analysis following crosslinker treatment. Due to the short revision window (10 days), these experiments could not be completed in time for the current resubmission. Results from these ongoing experiments will be incorporated into follow-up studies.

3. The authors report IL-13 and IL-16 as key inflammatory mediators, yet the role of IL-1β—typically central to NLRP3 activation—is less emphasized. Would a more detailed profiling of cytokine secretion, including IL-1β and IL-18, provide a more complete picture of the inflammatory cascade?

We thank the reviewer for the valuable insight and suggestion. In response, we performed a broader transcriptional analysis and generated a heatmap after data standardization (Z-score). The results have been included as Figure 4E in the revised manuscript.

Consistent with our earlier findings, EPA and UroA did not exhibit synergistic effects on IL-1β gene expression. However, we observed a synergistic reduction in a broader range of pro-inflammatory genes, including IL-6, TNFα, iNOS, MCP-1, and COX2. In addition, M2 macrophage biomarker genes (IL-4 and IL-13) showed slight increases in the EPA- and UroA-supplemented groups, with a more pronounced effect observed in response to UroA. Interestingly, IL-16 expression was not upregulated by either LPS or HS, suggesting that IL-16 may not play a direct role in LPS+HS-induced inflammasome activation in microglial cells.

4. While the study references physiologically relevant concentrations of EPA and UroA, it remains unclear whether these levels are achievable through dietary intake alone. Could the authors elaborate on the pharmacokinetics or bioavailability of these compounds in vivo, particularly in the context of HS or neuroinflammatory conditions?

We thank the reviewer for the valuable suggestion. In response, we have expanded the Discussion to provide a more detailed consideration of the brain bioavailability of EPA and UroA and clearly mentioned as a limitation of our study. Below is what we expanded in the Discussion Section.

Our present study has several limitations. The synergistic therapeutic mechanisms of EPA and UroA in the context of heat stroke (HS) have not been demonstrated in vivo, and their direct effects on neurons remain to be clarified. In addition, the brain bioavailability of these compounds represents an important gap. The concentrations used in our in vitro experiments (50 μM for EPA and 10 μM for UroA) are achievable in systemic circulation through dietary supplementation; however, their ability to cross the blood–brain barrier (BBB) is limited, with estimates suggesting that roughly one-tenth of circulating levels may reach brain tissue [59,60]. Both EPA and UroA can cross the BBB and undergo metabolic processing in the brain, which indicates potential involvement in neuroinflammatory regulation. Further in vivo studies are needed to clarify their relevance in central nervous system inflammation (Discussion line 399-402).

Citation

  1. Bazinet, R.P.; Metherel, A.H.; Chen, C.T.; Shaikh, S.R.; Nadjar, A.; Joffre, C.; Layé, S. Brain eicosapentaenoic acid metabolism as a lead for novel therapeutics in major depression. Brain, behavior, and immunity 2020, 85, 21-28, doi:10.1016/j.bbi.2019.07.001.
  2. Zhang, Q.; Zhang, W.; Yuan, X.; Peng, X.; Hu, G. Urolithin A in Central Nervous System Disorders: Therapeutic Applications and Challenges. Biomedicines 2025, 13, doi:10.3390/biomedicines13071553.

5.The proposed dietary implications are intriguing, especially regarding walnut-derived precursors. However, the translational leap from in vitro findings to dietary recommendations may benefit from further substantiation. Could the authors consider framing these suggestions more cautiously or referencing ongoing clinical or preclinical studies?

We fully understand and appreciate the reviewer’s concerns. In response to this comment, we have carefully revised the Discussion to present the potential effects of walnuts in a more balanced and cautious manner. Specifically, we have toned down speculative statements and reframed our interpretation to emphasize the preliminary nature of our findings. Rather than overstating the translational significance, we now highlight that while our results suggest a possible protective role of walnut supplementation, further validation through additional mechanistic studies and clinical investigations will be required (Discussion 394-397).

Please see the attached letter for the response to the reviewers

Round 2

Reviewer 1 Report

Comments and Suggestions for Authors

I am satisfied with the revisions and I recommend that the manuscript be accepted for publication.